# Ni-Pd-Incorporated Fe₃O₄ Yolk-Shelled Nanospheres as Efficient Magnetically Recyclable Catalysts for Reduction of N-Containing Unsaturated Compounds

**Dong Wang** [1], **Yi Li** [1], **Liangsong Wen** [1], **Jiangbo Xi** [1,*], **Pei Liu** [2,*], **Thomas Willum Hansen** [2] **and Ping Li** [3,*]

1. School of Chemistry and Environmental Engineering, Key Laboratory of Green Chemical Engineering Process of Ministry of Education, Engineering Research Center of Phosphorus Resources Development and Utilization of Ministry of Education, Hubei Key Laboratory of Novel Reactor and Green Chemical Technology, Key Laboratory of Novel Biomass-Based Environmental and Energy Materials in Petroleum and Chemical Industry, Wuhan Institute of Technology, Wuhan 430073, China
2. DTU Nanolab, Technical University of Denmark, Fysikvej, 2800 Kongens Lyngby, Denmark
3. School of Materials and Mechanical & Electrical Engineering, Jiangxi Key Laboratory of Surface Engineering, Jiangxi Science and Technology Normal University, Nanchang 330013, China
* Correspondence: jbxi@wit.edu.cn (J.X.); peiliu@dtu.dk (P.L.); lp1849065552@163.com (P.L.)

**Abstract:** The use of metal-based heterogeneous catalysts for the degradation of N-containing organic dyes has attracted much attention due to their excellent treatment efficiency and capability. Here, we report the synthesis of heterometals (Ni and Pd)-incorporated Fe₃O₄ (Ni-Pd/Fe₃O₄) yolk-shelled nanospheres for the catalytic reduction of N-containing organic dyes using a facile combination of solvothermal treatment and high-temperature annealing steps. Benefiting from the magnetic properties and the yolk-shelled structure of the Fe₃O₄ support, as well as the uniformly dispersed active heterometals incorporated in the shell and yolk of spherical Fe₃O₄ nanoparticles, the as-prepared Ni-Pd/Fe₃O₄ composite shows excellent recyclability and enhanced catalytic activity for three N-containing organic dyes (e.g., 4-nitrophenol, Congo red, and methyl orange) compared with its mono metal counterparts (e.g., Ni/Fe₃O₄ and Pd/Fe₃O₄). In the 4-nitrophenol reduction reaction, the catalytic activity of Ni-Pd/Fe₃O₄ was superior to many Fe₃O₄-supported nanocatalysts reported within the last five years. This work provides an effective strategy to boost the activity of iron oxide-based catalytic materials via dual or even multiple heterometallic incorporation strategy and sheds new light on environmental catalysis.

**Keywords:** Fe₃O₄ nanospheres; heterometal incorporation; magnetic catalyst; N-containing unsaturated compound; reduction reaction

## 1. Introduction

There are increasing concerns regarding the accumulation of organic pollutants such as N-containing dyes in the aquatic ecosystems [1]. These N-containing organic compounds are usually structurally stable and mostly refractory in the natural environment, thus threatening human health and the environment due to their intrinsic toxicity [2]. Therefore, effective techniques for dye-polluted wastewater treatment are highly desired but still need further investigation [3,4]. As commonly used dyes, N-containing organic compounds (e.g., nitroaromatics and azo compounds) usually have unsaturated chromophore groups, such as nitro- (O←N=O) and azo (-N=N-) moieties, along with aromatic rings in their structures. Considering that these unsaturated groups are reactive, it is possible to cleave the molecular structures of these toxic dyes by chemical hydrogenation or reduction reactions in the presence of an efficient catalyst [5]. In this way, N-containing organic dyes can be decolorized and converted into less harmful aminoaromatics [6], which are valuable intermediates for the industrial production of a variety of agrochemicals, pharmaceuticals, dyes, and pigments [7,8]. In recent years, the reduction of 4-nitrophenol, Congo red, and methyl

orange by sodium borohydride (NaBH$_4$) in aqueous solution has become an ubiquitous model reaction used to test catalyst activity [2,9,10]. However, it is still a daunting task to achieve high activity, selectivity, as well as recyclability for the catalyst during harsh reaction processes.

In recent years, the use of metal-based heterogeneous catalysts for decolorizing and converting N-containing organic dyes to aminoaromatic substances has attracted much attention due to their excellent treatment efficiency and capability [11–15]. In order to gain high catalytic efficiency and favorable utilization of active metal, the metal species are usually loaded on support materials in the form of nanoparticles [16], clusters [17], or even single atoms [18,19]. However, their small sizes lead to obvious drawbacks, such as difficulty in separating and recycling the catalyst from the reaction systems efficiently by conventional methods (e.g., filtration and centrifugation) [20]. To address this issue, scientists are trying to combine active components with magnetic materials. They introduced active metal species (e.g., nanoparticles, clusters, and single-atoms) to magnetic support materials and synthesized magnetically recyclable catalysts [21–23]. Due to the magnetic property of the catalysts, quick separation, and recycling of the catalysts from the reaction system can easily be achieved by applying a permanent magnet externally [24,25]. It is believed that magnetically recyclable catalysts are cost-effective and potentially applicable for industrial applications.

As a representative magnetic material, iron oxide (e.g., Fe$_3$O$_4$) is considered one of the promising candidates as a catalyst-supporting material due to its abundance and excellent stability [26]. Since the high specific surface area (SSA) of the support material is crucial to the exposure of active metal sites and mass transportation, this significantly influences the catalytic performance [27,28]. Incorporating active metal into the Fe$_3$O$_4$-based material with specific micro-/nano architectures should be an option for fabricating efficient and magnetically recyclable catalysts [29]. As a result, hierarchically structured magnetic Fe$_3$O$_4$ nanomaterials are ideal candidates as supports for catalysts [30]. However, it is still challenging to stably anchor high-density active metal sites on Fe$_3$O$_4$ materials due to the limited strength of the interaction between the iron oxide and heterometal species [31,32]. Therefore, Fe$_3$O$_4$ materials are usually functionalized via surface modification, such as coating with polymer [33]. For instance, Duan et al. fabricated an effective recyclable nanocatalyst based on double-shelled hollow nanospheres-supported Pd nanoparticles, in which magnetic Fe$_3$O$_4$ was functionalized with polydopamine [34]. Our previous work demonstrated that the incorporation of active metal into Fe$_3$O$_4$ was an effective and facile strategy to synthesize magnetic catalysts [35]. Moreover, findings indicated that dual or multiple metal-based catalysts exhibited superior catalytic performance to their mono metal-based counterparts [13,15,36–39]. Based on the above-mentioned discussions and catalyst design rationales, the incorporation of heteroatom metals into hierarchical Fe$_3$O$_4$ should be an efficient magnetically recyclable catalyst for the catalytic decolorization of N-containing organic dyes.

In this work, we report the synthesis of Ni and Pd-incorporated Fe$_3$O$_4$ (Ni-Pd/Fe$_3$O$_4$) yolk-shelled nanospheres via a combination of solvothermal treatment and high-temperature annealing. Benefiting from magnetic properties, the yolk-shelled structure, and uniformly dispersed active heterometals incorporated in the shell and yolk of spherical Fe$_3$O$_4$, the Ni-Pd/Fe$_3$O$_4$ composite showed excellent recyclability and enhanced catalytic activity for three N-containing organic dyes [e.g., 4-nitrophenol (4-NP), Congo red (CR), and methyl orange (MO)] compared with its mono metal counterparts (e.g., Ni/Fe$_3$O$_4$ and Pd/Fe$_3$O$_4$). Furthermore, the catalytic activity of Ni-Pd/Fe$_3$O$_4$ surpassed many Fe$_3$O$_4$-supported nanocatalysts reported within the last five years.

## 2. Results and Discussions

### 2.1. Preparation and Characterization of the Ni-Pd/Fe$_3$O$_4$ Catalyst

Ni and Pd-incorporated Fe$_3$O$_4$ (Ni-Pd/Fe$_3$O$_4$) spherical yolk-shelled nanocatalyst was synthesized via a modified combination method [40]. Fe(NO$_3$)$_3$·9H$_2$O and K$_2$PdCl$_4$ or

$NiCl_2 \cdot 6H_2O$ were used as metal precursors and dissolved in a mixture of deionized water, isopropanol, and glycerol (Figure 1). On the basis of our experimental observations in the present study and previous works [40,41], the formation of the yolk-shelled structure of Ni-Pd/$Fe_3O_4$ nanospheres could be explained by a self-templating mechanism. First, Fe, Ni and Pd ions coordinate with isopropanol to form Ni/Pd-incorporated Fe-isopropanol solid nanospheres in the solvothermal process. The resulting Ni/Pd-incorporated Fe-isopropanol solid nanospheres then gradually transform into a relatively thermodynamically stable Ni/Pd incorporated Fe-glycerate composite. During the solvothermal transformation process, Ni/Pd-incorporated Fe-glycerate grow on the surface of Ni/Pd incorporated Fe-isopropyl alcohol nanospheres at the expense of the gradual consumption of Ni/Pd-incorporated Fe-isopropanol. After the completion of this reaction, the Ni/Pd-incorporated Fe-isopropanol solid nanospheres partially convert into yolk-shelled nanospheres consisting of a Ni/Pd-incorporated Fe-glycerate shell and a Fe-isopropyl alcohol solid core. Finally, the obtained Ni/Pd-incorporated Fe-glycerate composite was annealed and transformed into a Ni-Pd/$Fe_3O_4$ yolk-shelled nanospherical catalyst.

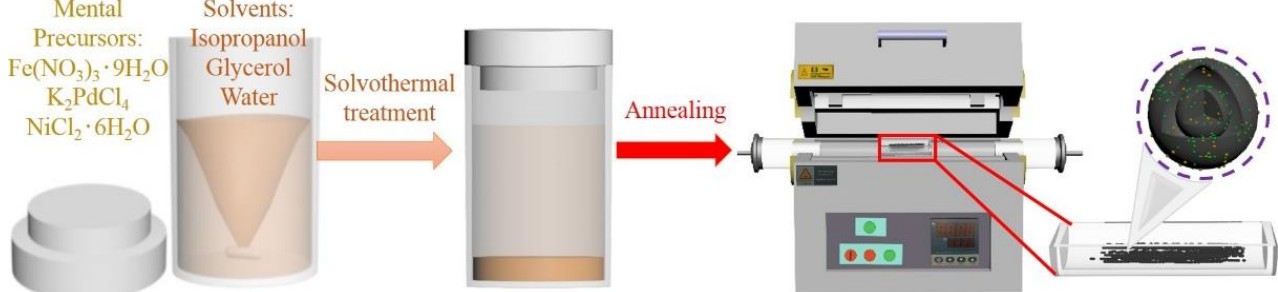

**Figure 1.** Schematic illustration of the preparation process of Ni-Pd/$Fe_3O_4$ catalyst.

The phase composition of the synthesized Ni-Pd/$Fe_3O_4$ catalyst was confirmed by X-ray diffraction (XRD). As illustrated in Figure 2a, the characteristic peaks at 18.3°, 30.1°, 35.5° 43.1°, 57.1°, and 62.6° match well with the (011), (112), (103), (004), (321), and (224) reflections of $Fe_3O_4$ (JCPDS No. 01-075-1609), respectively. We note the absence of metallic Ni and Pd peaks in the XRD patterns of the Ni-Pd/$Fe_3O_4$ sample, which should be attributed to the small size and low loading of Ni and Pd incorporated in the $Fe_3O_4$ support. In addition, the morphology of the synthesized Ni-Pd/$Fe_3O_4$ nanocatalysts was characterized by scanning electron microscopy (SEM). As can be seen from the SEM images, Ni-Pd/$Fe_3O_4$ presents a spherical nanostructure with a diameter of 500–800 nm (Figure 2b,c). Transmission electron microscopy (TEM) and aberration-corrected high-angle annular dark-field scanning transmission electron microscopy (HAADF-STEM) images further confirmed the spherical yolk-shelled structure. A yolk-like core was encapsulated in the thin shell for each Ni-Pd/$Fe_3O_4$ nanosphere (Figure 3a). The shell thickness of the Ni-Pd/$Fe_3O_4$ particles is about 60 nm, consisting of stacked tiny nanoparticles (Figure 3b,c), leading to the formation of a porous structure. In comparison, the pristine $Fe_3O_4$ nanospheres prepared without the Ni and Pd precursor show a hollow nanospherical structure (Figure S1). The STEM images and the energy dispersive X-ray (EDX) elemental mapping show that the Ni and Pd elements are uniformly distributed across the $Fe_3O_4$ nanospheres (Figure 3d–h). However, the crystallization degree of the $Fe_3O_4$ support is not high enough to discriminate Ni or Pd Fe species from the lattice of the $Fe_3O_4$ substrate (Figure S2). The SSA and porosity characteristics of the Ni-Pd/$Fe_3O_4$ hollow spherical nanocatalyst were analyzed by $N_2$ adsorption–desorption measurements and determined via Brunauer–Emmett–Teller (BET) and Barrett–Joyner–Halenda (BJH) methods. The H2-type $N_2$ adsorption–desorption isotherm exhibited a distinct hysteresis loop in the desorption branch. The SSA for the Ni-Pd/$Fe_3O_4$ was 140.4 $m^2$ $g^{-1}$, and the size of the pores (e.g., micropore and mesopore) mainly fall in the range from 1.5 to 10 nm (Figure 4a,b).

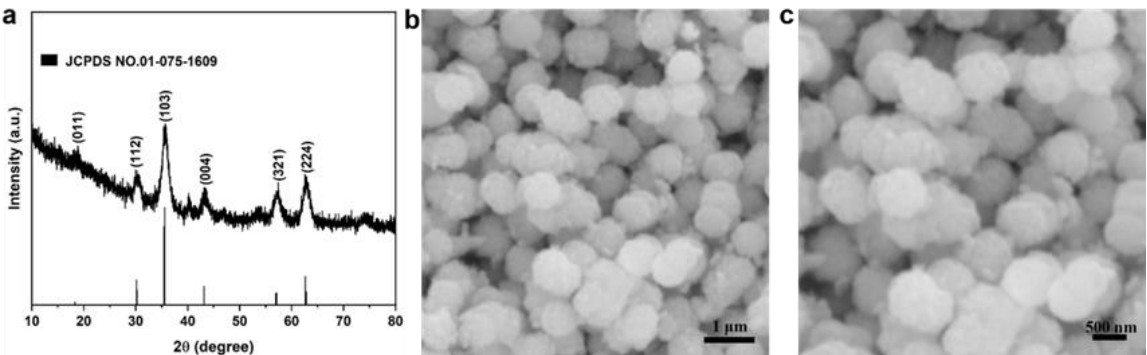

**Figure 2.** (**a**) XRD pattern of Ni-Pd/Fe$_3$O$_4$ and standard card JCPDS No. 01-075-1609. (**b**,**c**) SEM images of Ni-Pd/Fe$_3$O$_4$ yolk-shelled nanospheres.

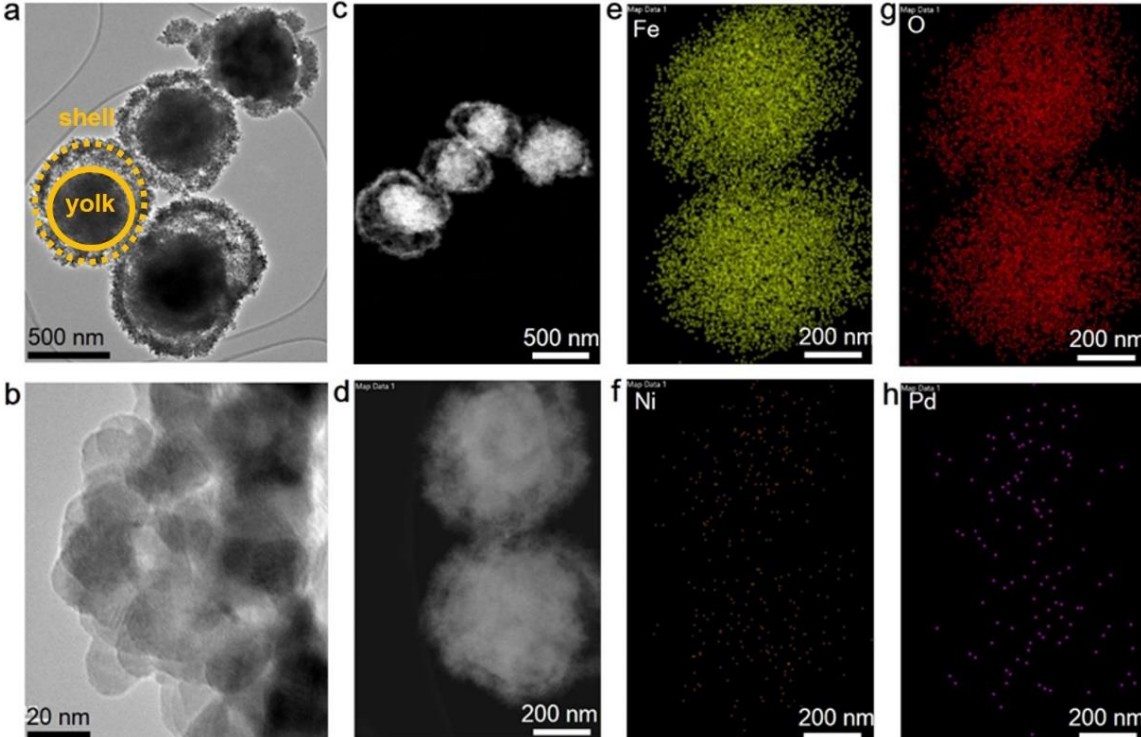

**Figure 3.** (**a**,**b**) TEM images, (**c**,**d**) HADDF-STEM images, and (**e**–**h**) corresponding EDX elemental mapping of Ni-Pd/Fe$_3$O$_4$ yolk-shelled nanospheres.

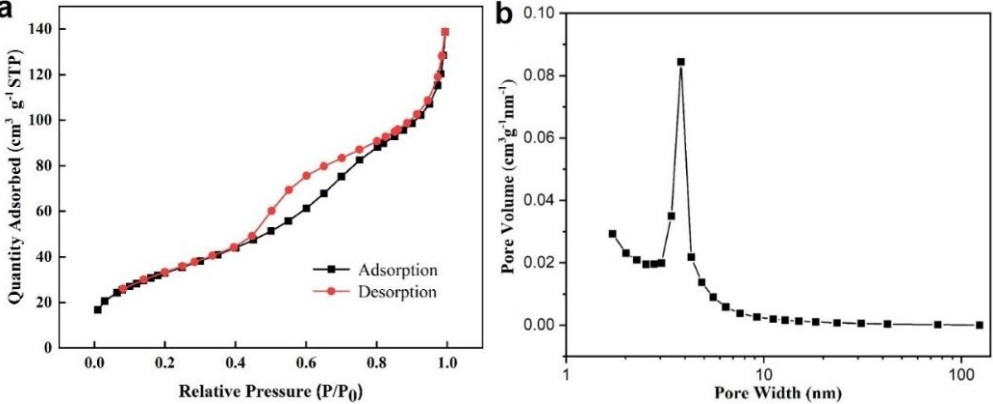

**Figure 4.** (**a**) N$_2$ adsorption–desorption isotherms and (**b**) pore size distribution for Ni-Pd/Fe$_3$O$_4$ catalyst.

X-ray photoelectron spectroscopy (XPS) analysis was conducted to determine the elemental composition and chemical state of the Ni-Pd/Fe$_3$O$_4$ nanocatalyst. As illustrated in Figure 5a, the survey spectra showed Fe, O, C, Ni, and Pd in the sample. The high-resolution spectra of Fe 2p exhibited two prominent peaks, which are assigned to Fe 2p$_{3/2}$ (at around 712.5 eV and 710.6 eV) and Fe 2p$_{1/2}$ (at 727.8 eV and 724.3 eV) of Fe$_3$O$_4$ (Figure 5b) [35]. The fitted doublet peak of the Pd 3d spectra centered at 335.6 eV and 341.0 eV assigned to the Pd(I) oxidation state (Figure 5c) [35,42]. The fitted Ni 2p spectra are shown in Figure 5d. The peaks at 855.7 eV and 862.3 eV are assigned to Ni 2p$_{3/2}$ and Ni 2p$_{1/2}$, respectively [43,44]. The loading of Ni and Pd in Ni-Pd/Fe$_3$O$_4$ were 1.34 wt.% and 0.90 wt.% respectively, as determined by inductively coupled plasma mass spectrometry (ICP-MS) measurement.

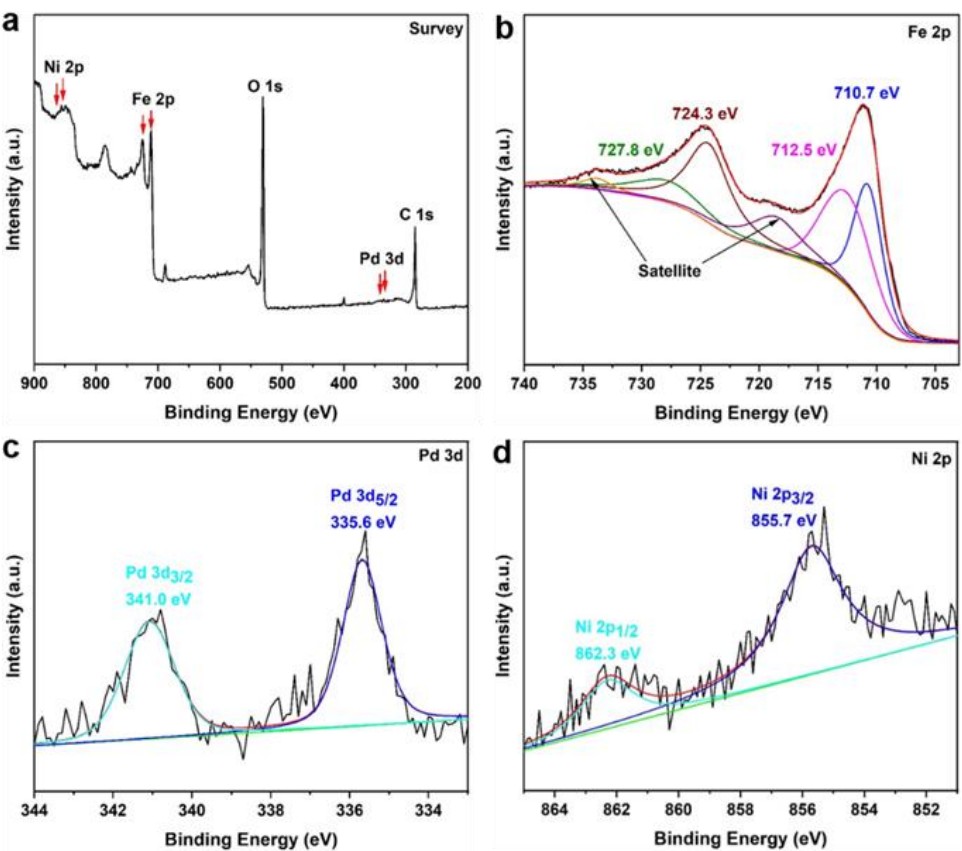

**Figure 5.** (**a**) XPS survey spectra of Ni-Pd/Fe$_3$O$_4$. High-resolution spectra of Fe 2p (**b**), Pd 3d (**c**), and Ni 2p (**d**).

### 2.2. Catalytic Performance of the Ni-Pd/Fe$_3$O$_4$ Catalyst

The catalytic performance of Ni-Pd/Fe$_3$O$_4$ catalyst towards the reduction of N-containing organic dyes (e.g., 4-NP, CR, and MO) was explored (Figure S3). Firstly, the catalytic efficiency of 4-NP reduction with NaBH$_4$ was investigated and quantitatively evaluated by turnover frequency (TOF), which was defined as the amount of 4-NP (mmol) converted into 4-AP per unit time catalyzed by per unit amount of active metal (mmol) [15].

$$\text{TOF}_{4-\text{NP}} = \frac{4 - \text{NP converted into } 4 - \text{AP (mmol)}}{\text{active Pd metal in the catalyst (mmol)} \times \text{time (min)}}$$

In order to probe the reaction kinetics, successive UV–vis detection of the reaction solution was conducted to monitor the reduction process. After adding the Ni-Pd/Fe$_3$O$_4$ catalyst into the mixture, the UV–vis absorbance peak of 4-NP-NaBH$_4$ at ca. 400 nm decreased quickly and the absorption peak of 4-AP at ca. 300 nm increased with time

(Figure 6a), indicating the successful conversion of 4-NP into 4-AP [45]. Accordingly, a gradual decolorization of the bright yellow aqueous 4-NP/NaBH$_4$ solution was observed during the catalytic process (4 min) (inset of Figure 6a). When the reaction is completed, Ni-Pd/Fe$_3$O$_4$ catalyst can be quickly magnetically separated from the aqueous reaction medium, allowing facile recycling of the catalyst (Figure S4). The Ni-Pd/Fe$_3$O$_4$ exhibited remarkable activity towards 4-NP reduction with a TOF as high as 295 min$^{-1}$, which was superior to its mono metal counterpart, such as Pd/Fe$_3$O$_4$ (TOF: 204 min$^{-1}$, Pd content: 0.87 wt.%) (Figures 7a and S5). We noted that Ni/Fe$_3$O$_4$ (Ni content: 1.16 wt.%) showed negligible catalytic activity for 4-NP reduction within 4 min (Figure S6), indicating a synergistic activity enhancement effect of the second heterometal (Ni). The O-containing Fe$_3$O$_4$ support could act as a ligand to stabilize the Ni and Pd species and facilitate their distribution. In addition, Ni could change the electronic structure of Pd and thus modulate the catalytic performance of the Ni-Pd/Fe$_3$O$_4$ catalyst [46,47]. It is worth noting that the catalytic activity of the Ni-Pd/Fe$_3$O$_4$ catalyst surpassed many of the Fe$_3$O$_4$-supported nanocatalysts reported within recent five years (Figure 7a and Table S1) [35,48–66]. In addition, the magnetic recyclability and durability were studied by repeating the catalytic reduction of 4-NP in the presence of a recycled Ni-Pd/Fe$_3$O$_4$ catalyst. Figure 6b shows the relationship between ln(A) (A denotes the absorbance at ca. 400 nm) and reaction time, the linear correlation indicated that the reduction process followed pseudo-first-order reaction kinetics. The apparent rate constant (K$_{app}$) was determined to be $1.51 \times 10^{-2}$ s$^{-1}$ from the slope of the linear correlation [67]. We calculated the ratio of rate constant K over the total weight of the catalyst, k = K$_{app}$/m. The activity factor, k, of the Ni-Pd/Fe$_3$O$_4$ catalyst was 15.1 s$^{-1}$ g$^{-1}$. As shown in Figure 7b, the conversion was nearly 100% on the eighth run and was maintained to 84.3% on the eleventh run, which indicated that Ni-Pd/Fe$_3$O$_4$ catalyst had excellent reusability and stability in the 4-NP reduction process. The morphology and phase composition of the recycled Ni-Pd/Fe$_3$O$_4$ were further characterized by SEM and XRD. We noted that a part of Ni-Pd/Fe$_3$O$_4$ yolk-shelled nanospheres broke after recycling from the reaction mixture (Figure S7). It may be hydrogen gas that was produced from NaBH$_4$ hydrolysis ejected from the void of the Ni-Pd/Fe$_3$O$_4$ hollow sphere and destroyed the yolk-shelled structure. XRD analysis of the recycled Pd-Fe$_3$O$_4$ catalyst showed the same diffraction peaks as the freshly prepared one, indicating that no obvious redox reaction occurred between Ni-Pd/Fe$_3$O$_4$ and NaBH$_4$ (Figure S8) and thus the magnetic property was maintained.

Ni-Pd/Fe$_3$O$_4$ catalyst also exhibited catalytic activity for the reduction of CR and MO, which are anionic azo dyes containing N=N bonds in their molecular structures. As shown in Figure 6c,e, 5 mL of aqueous CR or MO solution (5.0 mM) could be reduced entirely and decolorized when catalyzed by Ni-Pd/Fe$_3$O$_4$ catalyst using NaBH$_4$ as a reducing agent (57.0 mg) within 5 min or 40 min, respectively. We observed a time-dependent decrease in the UV–vis absorbance peak of CR ($\lambda_{max}$ = 494 nm) and MO ($\lambda_{max}$ = 464 nm) within 5 min and 40 min, respectively [68]. The complete disappearance of the peak confirmed the cleavage of N=N bonds and the formation of aminoaromatics. The corresponding K$_{app}$ values for CR and MO reduction reactions were determined as 7.85 s$^{-1}$ g$^{-1}$ and 0.76 s$^{-1}$ g$^{-1}$ (Figure 6d,f). The excellent catalytic performance (e.g., activity and recyclability) of Ni-Pd/Fe$_3$O$_4$ should arise from the uniformly incorporated dual heterometallic species on Fe$_3$O$_4$ support, the high SSA of Fe$_3$O$_4$ yolk-shelled nanosphere with a permeable porous shell, and its magnetically recyclable property.

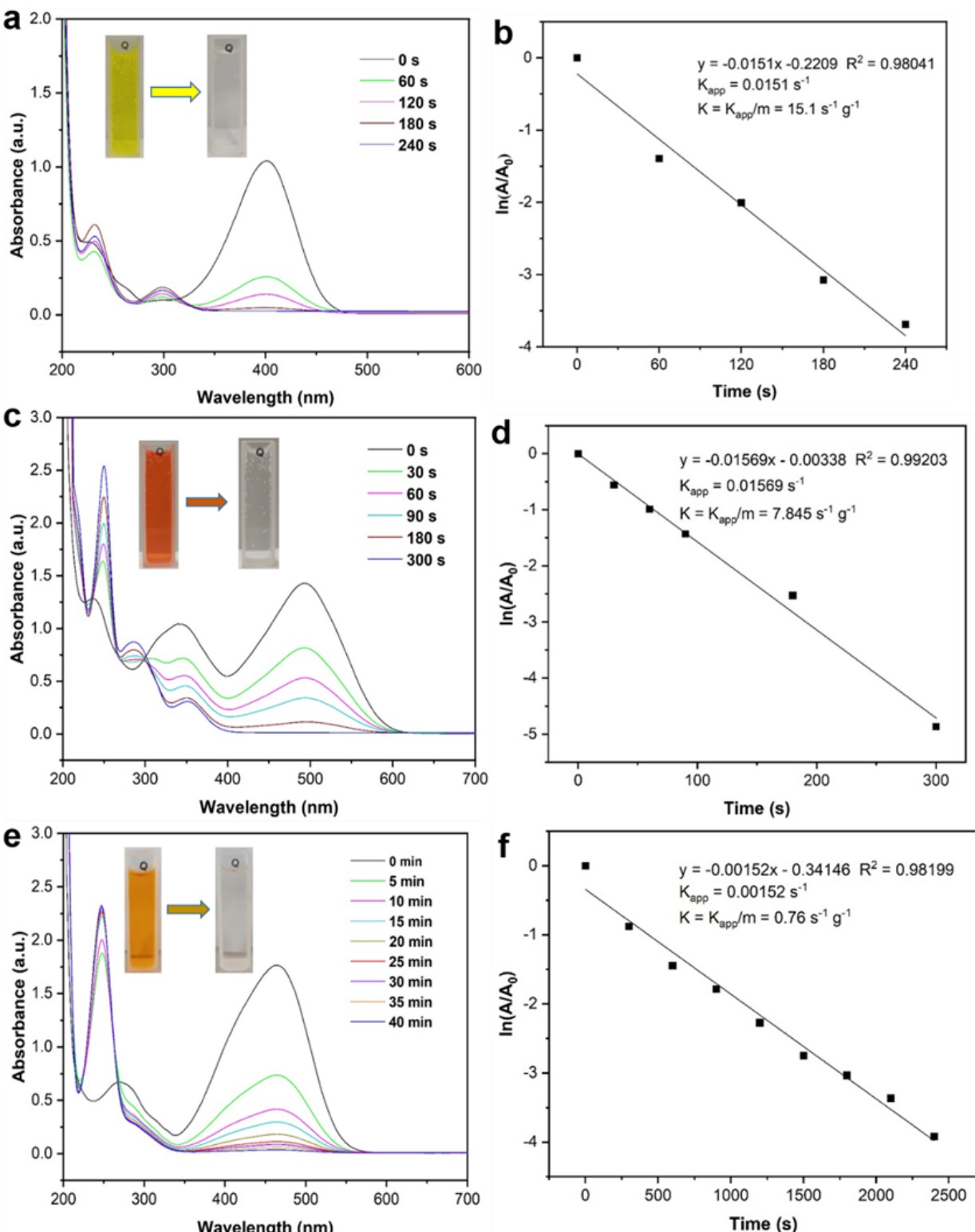

**Figure 6.** Time-dependent UV–vis spectra for (**a**) 4-NP, (**c**) CR, and (**e**) MO reduction by NaBH$_4$ in the presence of Ni-Pd/Fe$_3$O$_4$ catalyst. Insets in Figure 4b,c are the photographs showing the color fading of the reaction mixture. Rate constant versus time for (**b**) 4-NP, (**d**) CR, and (**f**) MO reduction by NaBH$_4$ in the presence of Ni-Pd/Fe$_3$O$_4$ catalyst.

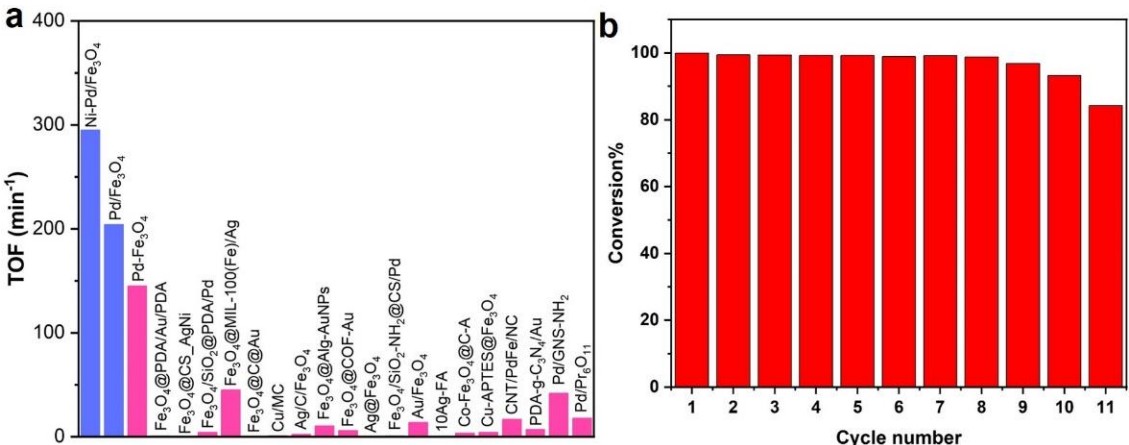

**Figure 7.** (**a**) The comparison of the catalytic activity (TOF) for the Ni-Pd/Fe$_3$O$_4$ catalyst and the Fe$_3$O$_4$ supported nanocatalysts reported within recent 5 years. Data adapted from Refs. [35,48–66] (**b**) Recycling of Ni-Pd/Fe$_3$O$_4$ catalyst for the 4-NP reduction by NaBH$_4$ in each cycle.

## 3. Materials and Methods

### 3.1. Materials Preparation

Preparation of Ni-Pd/Fe$_3$O$_4$ Yolk-Shelled Nanosphere

Ni-Pd/Fe$_3$O$_4$ yolk-shelled nanospheres were synthesized according to the modified solvothermal-annealing method reported previously [35,40]. Firstly, isopropanol (52.5 mL) and glycerol (7.5 mL) were successively added to a Teflon container (80 mL) and stirred to obtain a mixed solvent. Secondly, K$_2$PdCl$_4$ (1.7 mg) and NiCl$_2$·6H$_2$O (6.8 mg) were added to the mixture and stirred for 5 min to obtain a homogeneous mixture. Subsequently, Fe(NO$_3$)$_3$·9H$_2$O (202.0 mg) was added and stirred for another 5 min until all metal precursors were completely dissolved. After that, deionized water (1.0 mL) was injected into the above solution followed by additional stirring for 10 min. Subsequently, the Teflon container was sealed and transferred to a Teflon-lined stainless steel autoclave. After heating in an oven at 190 °C for 13 h, the Teflon container was naturally cooled to room temperature. A yellowish precipitate was obtained by centrifugation separation and subsequently washed with DI H$_2$O and ethanol three times. Drying at 60 °C for 6 h resulted in the intermediate Pd/Fe-glycerate sample. Finally, the Pd/Fe-glycerate was annealed at 350 °C for 3 h in a nitrogen atmosphere at a heating rate of 5 °C/min. The Fe$_3$O$_4$ hollow nanospheres were synthesized similarly without adding any metal precursor. The mono metal samples (Ni/Fe$_3$O$_4$ and Pd/Fe$_3$O$_4$) were synthesized similarly by adding only one active metal precursor (e.g., NiCl$_2$·6H$_2$O or K$_2$PdCl$_4$).

### 3.2. Catalytic Measurements of Ni-Pd/Fe$_3$O$_4$ Catalyst

3.2.1. Reduction of 4-NP, CR, and MO

For the 4-NP reduction reaction, 4-NP was first dissolved in 5 mL of water to form an aqueous 4-NP solution (20 mmol/L). NaBH$_4$ (378 mg) was then added into the aqueous 4-NP solution to obtain 4-NP-NaBH$_4$ mixture solution. After that, Ni-Pd/Fe$_3$O$_4$ catalyst (1.0 mg) was added into the mixture under vigorous stirring at ambient conditions (ca. 25 °C). The bright yellow 4-NP-NaBH$_4$ mixture faded gradually and finally became colorless, indicating complete conversion of 4-NP into 4-AP. The reaction process and conversion of 4-NP were continuously monitored by UV-vis measurements of the reaction mixture (Note: the reaction mixture should be filtrated to remove the catalyst and diluted to a moderate concentration before analysis). For comparison, the catalytic performance of Ni/Fe$_3$O$_4$ or Pd/Fe$_3$O$_4$ (1.0 mg) for 4-NP (3 mL, 20.0 mmol/L) reduction were conducted under similar reaction conditions. For CR and MO reduction reactions, Ni-Pd/Fe$_3$O$_4$ catalyst (2.0 mg) and 0.2 mL ethanol were subsequently added into 3 mL of CR or MO (5.0 mmol/L) and NaBH$_4$ (57.0 mg) aqueous solution under vigorous stirring at ambient



condition (ca. 25 °C). The conversion of CR and MO was continuously monitored by subjecting the filtrated reaction mixture to UV-vis measurements.

### 3.2.2. Recyclability or Durability Test

The durability of the $Ni-Pd/Fe_3O_4$ catalyst was examined by measuring the conversion with a constant reaction time. In a typical reaction run, $Ni-Pd/Fe_3O_4$ (5.0 mg) was added into the aqueous solution (5.0 mL) of 4-NP (20.0 mmol/L) and $NaBH_4$ (2.0 mol/L) with vigorous stirring under ambient conditions. The exact reaction time was considered as the constant reaction time for each catalytic run. The catalytic reduction process of the 4-NP was monitored by UV–vis spectroscopy analysis and color fading of the reaction mixture. The catalyst could be easily separated from the reaction mixture by a magnet. The recycled $Ni-Pd/Fe_3O_4$ was washed with water and ethanol and then used for the next run.

## 4. Conclusions

In summary, we report a highly efficient magnetically recyclable catalyst with heterometals (Ni and Pd) uniformly incorporated in $Fe_3O_4$ yolk-shelled nanospheres via solvothermal treatment and subsequent high-temperature annealing approaches. The high SSA, as well as the abundant mesopores on the spherical $Fe_3O_4$ shell, facilitated the exposure and accessibility of active sites and promoted mass transportation of reactants, and thus boosted the catalytic activity. The $Ni-Pd/Fe_3O_4$ catalyst showed excellent recyclability and high catalytic efficiency for the reduction of three N-containing organic dyes (e.g., 4-NP, CR, and MO) compared with its mono metal counterparts (e.g., $Ni/Fe_3O_4$ and $Pd/Fe_3O_4$). Furthermore, the kinetics of the catalytic reduction reaction were explored in detail. For the 4-NP reduction reaction, the catalytic efficiency of $Ni-Pd/Fe_3O_4$ surpassed that of many $Fe_3O_4$-supported nanocatalysts reported within the last five years. The present work provides a potential platform for designing and fabricating magnetically recyclable catalysts for various heterogeneous reactions.

**Supplementary Materials:** The following supporting information can be downloaded at: https://www.mdpi.com/article/10.3390/catal13010190/s1, Figure S1: SEM and TEM images; Figure S2: TEM image; Figure S3: reaction equations, Figure S4: photographs, Figure S5 and Figure S6: UV–vis spectra, Figure S7: SEM image; Figure S8: XRD pattern; Table S1: catalytic activity comparison of the prepared and previously reported catalysts.

**Author Contributions:** J.X., P.L. (Pei Liu) and P.L. (Ping Li) conceived and supervised the research. D.W., Y.L., and L.W. carried out the materials synthesis and the chemical catalysis. D.W., P.L. (Pei Liu), T.W.H. and Y.L. performed the materials characterizations. J.X., P.L. (Pei Liu), P.L. (Ping Li), and T.W.H. write, review, and edit the manuscript. All authors discussed the results and contributed to the manuscript. All authors have read and agreed to the published version of the manuscript.

**Funding:** This research was funded by the Key Research and Development Program of Hubei Province (grant number 2022BAA026), the Major project of Hubei Provincial Department of Education (grant number D20211502), the Postgraduate Innovation Foundation from Wuhan Institute of Technology (grant number CX2022436), the Open/Innovation Project of Key Laboratory of Novel Biomass-Based Environmental and Energy Materials in Petroleum and Chemical Industry (grant number 2022BEEA06), and the Open Project of Ministry-of-Education Key Laboratory for the Green Preparation and Application of Functional Materials and the Carlsberg Foundation (grant number CF20-0612).

**Data Availability Statement:** The authors confirm that the data supporting the findings of this study are available within the article. Derived data supporting the findings of this study are available on request.

**Conflicts of Interest:** The authors declare no conflict of interest.

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
