# Peer review of "Ni-Pd-Incorporated Fe3O4 Yolk-Shelled Nanospheres as Efficient Magnetically Recyclable Catalysts for Reduction of N-Containing Unsaturated Compounds"

_catalysts, doi:10.3390/catal13010190_

Round 1

Reviewer 1 Report

This manuscript reports the preparation of incorporated Fe3O4 yolk-shelled nanospheres through a two-step solvothermal/annealing method. The structure and composition of the catalyst were well characterized by SEM, TEM, HAADF-STEM, XRD, XPS, BET, ICP-MS, etc. The as-prepared catalyst exhibited an excellent catalytic performance toward the reduction of three N-containing organic dyes (e.g., 4-nitrophenol, Congo red and methyl orange). The results should be of interest to general readers. Therefore, this manuscript is recommended for publication after the following issues are properly addressed.

1.      Since the morphology of the catalysts is crucial to the performance, the advantages of the yolk-shelled morphology should be highlighted in the Introduction to help the readers to understand the structure-performance relationship.

2.      The EDS analysis revealed that Fe, O, Ni, and Pd species are rather uniformly distributed across the spherical nanoparticle. High-resolution TEM characterization is recommended to discriminate Pd or Ni species from Fe3O4.

3.      In order to confirm the stability of the yolk-shelled morphology, SEM images of recycled catalyst should be supplemented.

Author Response

We greatly appreciate you and the reviewers for your insightful comments, and have carefully revised the manuscript accordingly. The following is a detailed description of our responses to the questions raised by the reviewers’ concerns and the changes that we have made. We sincerely hope that the revised manuscript is now acceptable for publication in Catalysts. Your efforts in the review process of this manuscript are greatly appreciated. Please see enclosed a detail point-to-point response and revisions made in our revised manuscript.

Reviewer 2 Report

The authors synthesize heterometals (Ni and Pd) incorporated Fe3O4 (Ni-Pd/Fe3O4) yolk-shell nanospheres via a facile combination method consisting of solvothermal treatment and high-temperature annealing steps. The prepared species were characterized by XRD, SEM, EDX, XPS and N2 adsorption-desorption measurements. The obtained nanospheres composite shows excellent catalytic recyclability and enhanced catalytic activity for three N-containing organic dyes (e.g., 4-nitrophenol, Congo red and methyl orange) compared with its momometal counterparts (e.g., Ni/Fe3O4 and Pd/Fe3O4). This work is recommended for publication in Catalysts after minor modification.

  1. Authors should investigate the effect of the amount of the catalysts on the catalytic performance;

 2. Authors should investigate the effect of the interfering substances on the catalytic performance;

3. Authors should investigate the catalytic performance in real samples;

4. The catalysts should be characterized after the catalytic reaction in NaBH4 solution to confirm whether NaBH4 can reduce Fe3O4 to Fe2O3 or FeO resulting in the decomposition of Fe3O4.

Author Response

(The authors gave the same response as above.)

Reviewer 3 Report

The manuscript reported the synthesis and catalytic application of York-shell structured Fe3O4 nanospheres modified with heterometals. It shows excellent recyclability and enhanced catalytic activity for three N-based compounds. The manuscript can give some guidance for designing new catalysts for green chemistry. However, there are several points need to be addressed before publication:

1.       The yolk-shell structure was confusing. What’s the difference between the yolk and shell?

2.       What’s the formation mechanism of the claimed yolk-shell structure? What’s the structure without Ni and Pd precursor?

3.       Why did the author use these three types of N-contained compounds? Should state it in the introduction part.

4.       Figure 4b: consider the pore size distribution, were the N2 adsorption curve data or desorption curve data used for calculation? Should state that clearly. Also, according to IUPAC nomenclature, mesopore size is between 2 and 50 nm. Since the pore sizes were distributed from 1.5 nm to 10nm, the micropores should also exist in the achieved Ni-Pd/Fe3O4 except the mesopores.

5.       Ni-XPS spectra: it seems that there exists NiOx species at 853eV. Need to double check that.

6.       The manuscript said that Ni/Fe3O4 showed neglected catalytic activity for 4-NP reduction, but why it has no activity? Based on previous publication, the CuNi NPs have excellent properties for 4-NP reduction [MRS Advances 5 (27-28), 1491-1496. MRS Advances 4 (5-6), 263-269]. Is it due to the formation of Ni-Fe spinel NPs rather than Ni/Fe3O4?

7.       The author claims that the synergistic effect in Ni-Pd/Fe3O4 attributes to the enhancement of 4-NP reduction activity compared to that of Pd/Fe3O4. How about the ligand effect that Ni can adjust the Pd electronic structure and thus optimize the properties?

8.       Why were the loading amount of Ni (0.90wt%) and Pd (1.34%) in Fe3O4 chosen? Will the Pd:Ni ratio affect the catalytic properties?

Author Response

(The authors gave the same response as above.)
